# High-Fidelity Generalizable Neural Surface Reconstruction with Sparse Scene Representations

## Abstract

Generalizable neural surface reconstruction has become a compelling technique at reconstructing 3D scenes from sparse input views without per-scene optimization. In these methods, dense 3D feature volumes have proven very effective as a global scene representation. Unfortunately, this representation severely limits their high-resolution modeling abilities and reconstruction accuraciy because memory requirements scale cubically with voxel resolution. In this paper, we propose a novel sparse-representation approach that dramatically improves memory efficiency and allows for more accurate surface reconstructions. Our method employs a two-stage pipeline: We first train a neural network to predict voxel occupancy probabilities from the given posed images, then we restrict feature computation and volume rendering to the sparse voxels with sufficiently high occupancy estimates. To support this sparse representation, we develop specialized algorithms for efficient sampling, feature aggregation, and spatial querying that overcome the dense-volume assumptions of existing approaches. Extensive experiments on standard benchmarks demonstrate that our sparse representation enables scene reconstruction at a $512^3$ resolution, compared to the typical $128^3$ resolution possible with existing methods on similar hardware. We also achieve superior reconstruction accuracy compared to current state-of-the-art approaches. Our work establishes sparse neural representations as a promising direction for scalable, high-quality 3D reconstruction.

## 1 Introduction

The emergence of neural implicit representation techniques, starting with the seminal Neural Radiance Fields (NeRF) (Mildenhall et al., 2020) and continuing with Neural Implicit Surfaces (NeuS) (Yariv et al., 2020; Wang et al., 2021a; Yariv et al., 2021; Li et al., 2023), has greatly boosted novel view synthesis and multi-view geometry reconstruction. These implicit surface reconstruction algorithms take posed images as input, and optimize a Signed Distance Function (SDF) to minimize a volume rendering loss, which yields accurate 3D reconstructions given enough views. A major drawback, however, is that the optimization must be performed from scratch for every new set of views, which makes these approaches computationally demanding. Furthermore, accuracy tends to decline when only a few views are available. Newer 3D Gaussian Splatting-based methods (Kerbl et al., 2023) are subject to the same limitations (Huang et al., 2024; Yu et al., 2024).

Generalizable approaches to Neural Surface Reconstruction (GNSRs) (Liang et al., 2024; Ren et al., 2023; Xu et al., 2023; Long et al., 2022; Na et al., 2024; Younes et al., 2024) are designed to remove these restrictions. These methods are pre-trained on a large number of scenes. As a result, the rendering of a novel scene simply becomes a test-time inference procedure, without any optimization required. Volume rendering and scene reconstruction are performed by relying on a 3D scene representation produced by a neural network that takes as input the images. This representation is typically in the form of a dense volume of high-dimensional features. However, even though they are much faster, these methods still have yet to achieve their full potential because the necessary volumetric feature representation is extremely memory-intensive. This drastically constrains their operational resolution as shown in Tab. 1, and adversely impacts reconstruction quality and detail preservation.

| Multi-view Images | Occupancy Field @ $128^3$ | Supersampled @ $512^3$ | Reconstructed Surface |

Figure 1: **Two-Stage Approach**. From multiple images, we predict occupancy field that is then supersampled to build a sparse scene representation. This lets us reconstruct high-fidelity surfaces by operating at previously unattainable $512^3$ resolution.

Sparse scene representation is a natural answer to this problem. Since surfaces are mathematically of measure zero, they occupy only a small fraction of voxels in a discretized 3D space, which may be leveraged for high-resolution, memory-efficient reconstruction. However, this brings significant challenges. Even though there has been work in that direction, no truly satisfactory approach exists yet. For example, SparseNeuS (Long et al., 2022) uses a coarse-to-fine scheme but still has to query a densified feature volumes, which negates the benefit of doing so.

In this paper, we propose a novel and effective way of bringing sparsity into GNSRs. Specifically, we introduce a nested two-stage architecture that enables our GNSR to operate at significantly higher resolutions than previous ones, while maintaining both generalizability across scenes and the ability to handle cases where only a limited number of views are available. As illustrated by Fig. 1, the two stages perform the following tasks:

1. **Computing occupancy at low resolution.** Voxels from a coarse grid are labeled as containing a surface element or not, by training a network using posed images as input. To avoid missing out parts of the surface, we implemented this so as to avoid false negatives.

2. **Computing sparse feature representations at a high resolution.** High-dimensional features are constructed at a higher-resolution but only within the occupied voxels, where volume rendering can be performed. This allows for fine-grained neural surface reconstruction while keeping memory and computational costs at a minimum.

Fig. 2 summarizes our *Sparse Volumetric Reconstruction* approach, which we dub *SVRecon*. While it is is conceptually simple, the new method requires rethinking and reformulating the algorithms used by existing methods, which are designed for handling dense representations. To this end, we develop new algorithms dedicated to handling sparse feature volumes to perform ray sampling, feature aggregation, and query operations. These technical innovations are key to realizing the theoretical memory advantages of our approach while maintaining computational efficiency.

| Methods | $128^3$ | $192^3$ | $256^3$ | $512^3$ |
|---|---|---|---|---|
| SparseNeuS (Long et al., 2022) | ✓ | ✓ | ✗ | ✗ |
| VolRecon (Ren et al., 2023) | ✓ | ✗ | ✗ | ✗ |
| ReTR (Liang et al., 2024) | ✓ | ✗ | ✗ | ✗ |
| Ours (*SVRecon*) | ✓ | ✓ | ✓ | ✓ |

Table 1: **Resolutions handled by GNSR algorithms at training time**. Crosses indicate that the memory requirements were too large on a 32GB NVIDIA Tesla V100 GPU, with the same setting of batch size 1 and 1024 sampled rays for all methods.

Testing on public datasets demonstrates that our sparse-representation method is highly effective, enabling us to operate at a high resolution of $512^3$ on standard hardware with 32GB of VRAM, thus improving the final reconstruction accuracy beyond the state-of-the art.

## 2 RELATED WORKS

**Multi-View Stereo.** Multi-view stereo (MVS) has long been known to be effective for 3D reconstruction (Fua, 1997; Hartley & Zisserman, 2000; Shum & Kang, 2000; Seitz et al., 2006; Furukawa & Ponce, 2009; Lhuillier & Quan, 2005; Kostrikov & Gall, 2014; Kutulakos & Seitz, 2000). Modern multi-view stereo methods can be categorized into depth-map ones (Yao et al., 2018; 2019; Gu et al., 2020; Wang et al., 2022; Ding et al., 2022; Cao et al., 2024) and volumetric ones (Ji et al., 2017; 2020; Kar et al., 2017). Depth-map methods rely on view-dependent frustums as an indirect model of 3D space, which limits the reconstruction accuracy without providing novel view rendering capabilities. Volumetric methods are directly related to our method, as they also operate on a

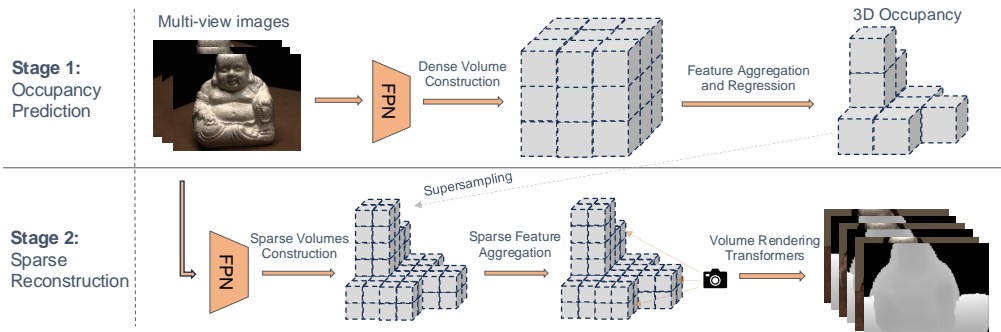

Figure 2: $SVRecon$ **pipeline.** First, we build a low-resolution dense 3D feature volume from multi-view features and predict 3D occupancies. Second, we construct high-resolution sparse feature volumes where the predicted occupancy warrants it. Finally, volume rendering Transformers are used to aggregate per-ray sampled features and infer color and depth, enabling scene reconstruction.

volumetric 3D space. Comparatively, our scene representation and volume rendering formulation makes it possible to recover finer details.

**Neural Scene Reconstruction.** The success of NeRFs (Mildenhall et al., 2020) has prompted researchers to examine the use of Neural Implicit Representations for surface reconstruction as an alternative to MVS. These line of works includes IDR (Yariv et al., 2020), VolSDF (Yariv et al., 2021), NeuS (Wang et al., 2021a), and Neuralangelo (Li et al., 2023), which are able to recover fine geometry given dense input views. More recently, SparseCraft (Younes et al., 2024) proposes to use stereopsis cues regularization to enhance surface reconstruction qualities under sparse input views. However, all these methods are not generalizable and need to be trained per scene.

**Generalizable Surface Reconstruction.** Many generalizable approaches have been proposed to dispense with intensive per-scene optimization. These methods include volumetric methods such as SparseNeuS (Long et al., 2022), VolRecon (Ren et al., 2023), ReTR (Liang et al., 2024); and frustum-based 3D representations (Na et al., 2024; Xu et al., 2023). However, all of these methods have to balance reconstruction quality against storage overhead due to the cost of their global scene representations, which our method improves upon. We note that SparseNeuS bears some similarities to our method in its coarse-to-fine scheme. However, their formulation still relies on densified feature volumes and thus cannot be seen as a sparse representation, their method limitations are also identified in our experiment by direct comparison. Please refer to Tab. 1 for a detailed comparison of admissible resolutions of different methods.

**Gaussian Splatting.** Gaussian splatting (Kerbl et al., 2023) have emerged as a powerful alternative to NeRF methods for novel view synthesis. The same trend is at work for 3D reconstruction, with the advent of methods such as 2D Gaussian Splatting (Huang et al., 2024) and Gaussian Opacity Fields (Yu et al., 2024). There are also recent attempts at making Gaussian Splatting generalizable (Zhang et al., 2025). However, generalizable surface reconstruction is still an open problem for Gaussian Splatting based methods due to the inherent difficulty to model surfaces with Gaussians.

## 3 BACKGROUND: GENERALIZABLE NERFS

We now briefly review the general formulation of the original and generalizable NeRFs. In both cases, the goal is to reconstruct a scene captured by a set of posed images $\mathcal{I} = \{(\mathbf{I}_i, \pi_i)\}_{i=1}^{M}$, where $\mathbf{I}_i \in \mathbb{R}^{H \times W \times 3}$ and $\pi_i \in \mathbb{R}^{3 \times 4}$ denote the projection matrix associated with the $i$-th view.

Given $\mathcal{I}$ as input, the aim is to learn a scene representation function F and a rendering function R

$$\mathsf{F} : (\mathbf{x}, \mathbf{d}) \rightarrow \mathbf{f}, \tag{1}$$

$$\mathsf{R} : \{(\mathbf{x}_i, \mathbf{f}_i)\}_{i=1}^{N} \rightarrow (C, D), \tag{2}$$

where F maps a 3D location $\mathbf{x} \in \mathbb{R}^3$ and viewing direction $\mathbf{d} \in \mathbb{S}^2$ to a feature vector $\mathbf{f} \in \mathbb{R}^C$, and R is a function that takes ray-sampled feature vectors as input and outputs a rendered color $C$ and optionally a depth value $D$.

In the original NeRF, $\mathbf{f}$ directly encodes an emitted color and a volume density, R is a handcrafted volume rendering function. The scene representation function F is implemented by a network with

weights $\lambda$. Learning them requires per-scene minimization of the photometric loss

$$\arg\min_{\lambda} \sum_{C_i \in \mathcal{I}} \mathcal{L}(C_i - \hat{C}_i) \,, \qquad (3)$$

where $\mathcal{L}$ is usually taken to be square norm.

In a generalizable NeRF, the feature vector $\mathbf{f}$ is high-dimensional and does not have physical meanings. The handcrafted rendering function R is replaced by one implemented by a network with weights $\phi$, while F uses a different architecture that allows for generalization and has weights $\theta$. The weights $\theta$ and $\phi$ are learned by training on a large corpus of scenes $\mathcal{D} = \{(\mathcal{I}_i, M_i)\}_{i=1}^{N_t}$, where $M_i$ denotes the ground-truth depth map corresponding to image $\mathcal{I}_i$. This involves minimizing

$$\arg\min_{\theta, \phi} \sum_{C_i, D_i \in \mathcal{D}} \mathcal{L}(C_i - \hat{C}_i) + \mathcal{L}(D_i - \hat{D}_i) \,. \qquad (4)$$

The key advantage over a vanilla NeRF is that, given the novel scene $\mathcal{I}$, scene representation function F can be obtained via direct inference without the need of per-scene optimization as in Eq. 3. We adopt this general formulation for our method. The main drawback is that having to store a dense feature volume as scene representation and having to perform computations on it is memory intensive, hence the resolution limitations of Tab. 1.

## 4 METHODOLOGY

We now present our $SVRecon$ approach to generalizable NeRFs that requires far less memory and can therefore be handle larger resolutions. Our main contribution lies in replacing the scene representation function F of Eq. 1 by one that produces a sparse representation and that we denote as $\mathsf{S} : (\mathbf{x}, \mathbf{d}) \to \mathbf{f}$. As discussed in the introduction, S operates in two stages. It first estimates occupancy in a low resolution—-typically $128^3$—-volume. The occupied voxels are then supersampled to a higher resolution, typically $512^3$, and the high-dimensional features $\mathbf{f}$ are computed within the resulting high-resolution voxels, as depicted by Fig. 2. The sparse representation creates challenges in volume rendering because the ray sampling, querying, and feature aggregations have to be redesigned to handle sparse volumes instead of dense ones, which will also be discussed.

Note that, in theory at least, instead of adopting the simple occupancy prediction described above, we could have used a more sophisticated octree-like structure, a popular choice to represent 3D geometry sparsely. However, the variable-resolution structure of octrees would have made volume rendering even more complex for no obvious gain.

In the remainder of this section, we first introduce the representation we use in both stages for individual voxels and the corresponding features. We then present the two stages to compute our sparse scene representation S. Finally, we discuss the learning of the $\theta$ and $\phi$ parameters by minimizing Eq. 4 given such a sparse representation, under a volume rendering framework.

### 4.1 REPRESENTING VOXELS AND FEATURES

At low-resolution we use a dense volume while we use a sparse one at high-resolution. Yet, we represent voxels in the same way in both cases. Given the center point $\mathbf{x}$ of a voxel, we first project it onto the images to obtain $M$-view features

$$\{\mathbf{f}_{I_i}(\pi_i(\mathbf{x}))\}_{i=1}^{M}, \qquad (5)$$

by bilinear interpolation, where $\mathbf{f}_{I_i}$ denotes image features computed by using the Feature Pyramid Network (Lin et al., 2017a). We then write the feature vector associated to $\mathbf{x}$ as

$$\mathbf{V}(\mathbf{x}) = \mathrm{MeanVar}(\{\mathbf{f}_{I_i}(\pi_i(\mathbf{x}))\}_{i=1}^{M}) \,, \qquad (6)$$

where $\mathrm{MeanVar}$ denotes the concatenation of per-channel mean and variance computed from the $M$-view features, and $\mathbf{V}$ denotes the raw dense/sparse feature volumes before a 3D global aggregation process. We take the dimension of $\mathbf{V}(\mathbf{x})$ to be the base feature channel $C_f$.

### 4.2 SPARSE SCENE REPRESENTATION

We now described the two stages—first occupancy prediction and then supersampling the occupied voxels—involved in building a sparse scene representation.

**Occupancy Prediction** Assuming that a scene bounding box is given, we voxelize it using a pre-defined voxel resolution. Due to memory constraints, the resolution cannot be too high. We compute the feature vector using Eq. 6 for each voxel, resulting in a dense feature volume $\mathbf{V}_d \in \mathbb{R}^{C \times K^3}$, where $K$ denotes the resolution. We then use a 3D U-Net (Ronneberger et al., 2015) $\Psi$ followed by a linear regression head $\mathfrak{R}$ to go from raw features $\mathbf{V}_d$ to the occupancy prediction $\mathbf{O} \in \mathbb{R}^{K^3}$. We write

$$\mathbf{O} = \mathfrak{R}(\Psi(\mathbf{V}_d)) \tag{7}$$

To train $\Psi$ and $\mathfrak{R}$, we rely on the ground-truth surfaces to estimate ground-truth occupancies $\mathbf{O}^{gt}$. In practice, we generate point clouds by merging the ground truth depth maps of each input view. This works better than using the ground-truth point cloud of the scene, which may contain points that are occluded in the input views and thereby impossible to predict. Since the overwhelming majority of voxels are empty in 3D space, we minimize a focal loss (Lin et al., 2017b) with focusing parameter $\gamma = 2$ during training to counteract the large class-imbalance. We take it to be

$$\mathcal{L}_{fc} = -\sum_{i,j,k} (1 - p_{ijk})^\gamma \log(p_{ijk}) \tag{8}$$

where

$$p_{ijk} = \begin{cases} \mathbf{O}_{ijk} & \text{if } \mathbf{O}_{ijk}^{gt} = 1 \\ 1 - \mathbf{O}_{ijk} & \text{if } \mathbf{O}_{ijk}^{gt} = 0 \end{cases}$$

At inference time, we simply threshold the occupancy predictions and apply a dilation process to yield the final binary predictions $\widetilde{\mathbf{O}} = \text{dilate}(\mathbb{I}(\mathbf{O} \geq \tau))$. The predicted occupancy is critical to the performance of our method. We prioritize recall in our method by using a small $\tau$ and employing a dilation process to ensure least geometry loss. See Supp. A.2 for more details.

**Supersampling.** The output $\widetilde{\mathbf{O}} \in \mathbb{R}^{K^3}$ is a low-resolution occupancy field. For scene representation purposes, we *supersample* the occupied voxels by increasing the resolution of them while ignoring the empty ones. Specifically, for a given voxel, the supersampling operation is performed by placing $s \times s \times s$ regular-grid samples within it, lifting the resolution from $K^3$ to $(sK)^3$. After supersampling, each voxel will be divided into mini-voxels, which we take and follow Eq. 6 to construct volumetric representations, resulting in raw 3D sparse feature volumes $\mathbf{V}_s \in \mathbb{R}^{N \times C \times s^3}$, where $N$ denotes the number of occupied voxels. We denote each $s \times s \times s$ block as a mini-volume.

We use the SparseUNet implementation from the sparse convolution library `torchsparse` (Tang et al., 2023) to perform 3D feature aggregation, as shown in the middle of the bottom row of Fig. 2. The sparse scene representation $\mathsf{S}$ is obtained as $\mathbf{S} = \Psi(\mathbf{V}_s)$ , where $\mathbf{S} \in \mathbb{R}^{N \times C \times s^3}$ and $\Psi$ denotes the 3D SparseUNet.

### 4.3 SPARSE VOLUMETRIC RECONSTRUCTION

Minimizing the losses of Eq. 4, requires repeating evaluating R given sparse scene representation $\mathsf{S}$, which means that R must handle efficiently the sparse nature of our representation. This means that the mechanisms for ray sampling. querying volume locations, and volume rendering used by R have to be redesigned to handle sparse volumes instead of dense ones.

**Ray Sampling.** Sampling along the ray has to be adapted because the only meaningful samples are those within occupied voxels. For any given ray, we detect its intersections with every occupied voxel and confine the sampling to ray fragments within occupied ones. We take an arbitrary sampled ray to be $\mathbf{r}(t_i) = \mathbf{o} + t_i \mathbf{d}$, $i = 1, 2, ..., N_s$, where $t_i$ denotes the samples, $\mathbf{o}$ and $\mathbf{d}$ denote ray origin and ray direction respectively.

**Querying S.** The volume rendering process is predicated on the ability to continuously query at arbitrary continuous locations along 3D rays, which is easy to achieve in a dense volume but not sparse ones due to their irregularity. Here we propose a simple yet effective algorithm to query the sparse feature volumes efficiently as shown in Alg. 1. The key to make our query algorithm very efficient is our specific data struture: our sparse volumes are a collection of small regular volumes spanning the occupied voxels. We pre-compute a dense lookup table $\mathbf{H}$ to encode the order of stored small volumes in memory and then perform the operations below. Please refer to Supp. A.1 for more details.

---

**Algorithm 1** Querying Sparse Volumes

1: **Input:** Querying point $\mathbf{x}$, lookup table $\mathbf{H} \in K^3$, sparse feature volumes $\mathbf{S} \in \mathbb{R}^{N \times s^3 \times C}$.
2: Find corner points surrounding $\mathbf{x}$, denote their coordinates $\{\mathbf{v}_g^i\}_{i=1}^8$ at scale $(sK)^3$.
3: Convert $\{\mathbf{v}_g^i\}_{i=1}^8$ at scale $(sK)^3$ to $\{\mathbf{v}_o^i\}_{i=1}^8$ at scale $K^3$ plus local shift $\{\mathbf{v}_l^i\}_{i=1}^8$ at scale $s^3$.
4: Get corner features from memory by $\mathbf{S}[\mathbf{H}[\mathbf{v}_o^i], \mathbf{v}_l^i, :]$.
5: Perform trilinear interpolation with corner features.
6: **Output:** Interpolated feature $\mathbf{f}_{vol}$.

---

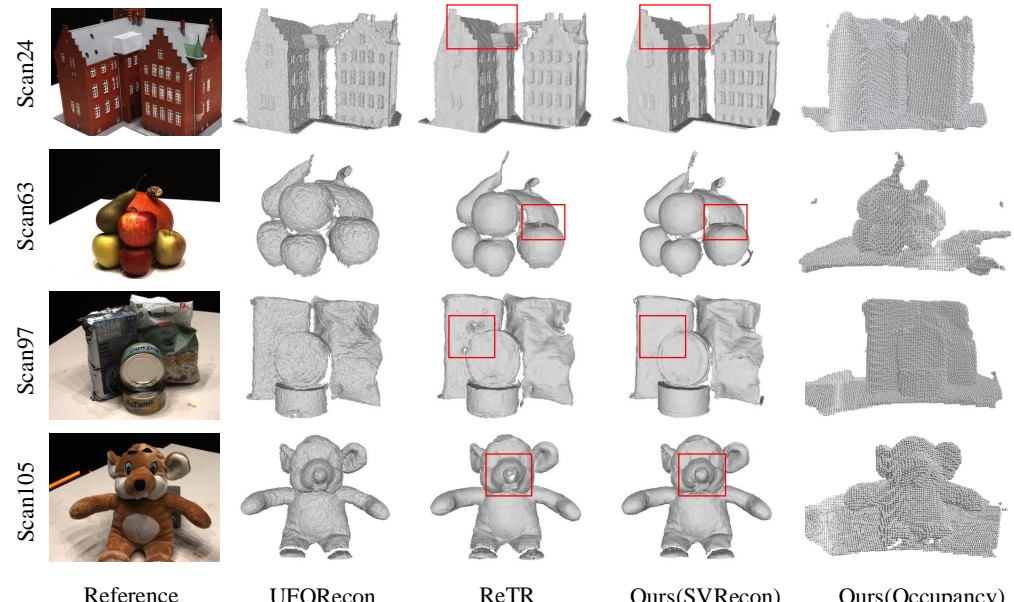

Figure 3: **Sparse-view surface reconstructions on DTU test scenes.** In the middle columns, we show shaded surfaces for UFORecon, ReTR, and our approach. In the rightmost column, we show the occupancies predicted by the first stage of our method at resolution $128^3$. Voxels are shrunk slightly for visualization purposes. The red rectangles highlight the region with visible differences.

**Rendering Function** R. We adopt the generalized volume rendering equation of (Liang et al., 2024) to model R. Specifically, the augmented feature representation for a ray sample $t_i$ is $\mathbf{f}_i^r = \mathrm{cat}(\mathbf{f}_{vol}, \mathbf{f}_{proj}, \beta)$, where $\beta$ denotes positional encoding (Wang et al., 2021b), $\mathrm{cat}(\cdot)$ denotes concatenation, $\mathbf{f}_{vol}$ denotes queried feature from $\mathbf{S}$, and $\mathbf{f}_{proj}$ denotes the aggregated projection features of Eq. 5 using a Transformer network (Ren et al., 2023). We write

$$C(\mathbf{r}), D(\mathbf{r}) = \mathcal{C}\left(\sum_{i=1}^{N_s} \sigma\left(\frac{q(\mathbf{f}^{tok})k(\mathbf{f}_i^{occ})^\top}{\sqrt{D}}\right)v(\mathbf{f}_i^r)\right), \sum_{i=1}^{N} \sigma\left(\frac{q(\mathbf{f}^{tok})k(\mathbf{f}_i^r)^\top}{\sqrt{D}}\right)t_i \qquad (9)$$

where $\sigma$ denotes the *softmax* operation, $\mathbf{f}^{tok}$ is a learnable token, $\mathbf{f}_i^{occ}$ denotes occlusion-aware feature derived from $\mathbf{f}_i^r$, $q(\cdot), k(\cdot), v(\cdot)$ are linear layers, and $\mathcal{C}(\cdot)$ is an MLP. To learn the network weights, we minimize the loss function

$$\mathcal{L} = \mathcal{L}_{\text{color}} + \alpha\mathcal{L}_{\text{depth}} = \frac{1}{N_s}\sum_{i=1}^{N_s}\|C(\mathbf{r}) - C_g(\mathbf{r})\|_2 + \alpha\frac{1}{N_d}\sum_{i=1}^{N_d}|D(\mathbf{r}) - D_g(\mathbf{r})|, \qquad (10)$$

where $C_g(\mathbf{r})$ and $D_g(\mathbf{r})$ are ground truth color and depth, respectively, $\alpha$ denotes a weight coefficient to balance the two terms, $N_s$ the number of sampled rays and $N_d$ the number of rays with valid depth.

| Scan | Mean (CD)↓ | 24 | 37 | 40 | 55 | 63 | 65 | 69 | 83 | 97 | 105 | 106 | 110 | 114 | 118 | 122 |
|---|---|---|---|---|---|---|---|---|---|---|---|---|---|---|---|---|
| COLMAP | 1.52 | 0.90 | 2.89 | 1.63 | 1.08 | 2.18 | 1.94 | 1.61 | 1.30 | 2.34 | 1.28 | 1.10 | 1.42 | 0.76 | 1.17 | 1.14 |
| TransMVSNet | 1.35 | 1.07 | 3.14 | 2.39 | 1.30 | 1.35 | 1.61 | 0.73 | 1.60 | 1.15 | 0.94 | 1.34 | **0.46** | 0.60 | 1.20 | 1.46 |
| VolSDF | 3.41 | 4.03 | 4.21 | 6.12 | 1.63 | 3.24 | 2.73 | 2.84 | 1.63 | 5.14 | 3.09 | 2.08 | 4.81 | 0.60 | 3.51 | 2.18 |
| NeuS | 4.00 | 4.57 | 4.49 | 3.97 | 4.32 | 4.63 | 1.95 | 4.68 | 3.83 | 4.15 | 2.50 | 1.52 | 6.47 | 1.26 | 5.57 | 6.11 |
| SparseNeuS-ft | 1.27 | 1.29 | 2.27 | 1.57 | 0.88 | 1.61 | 1.86 | 1.06 | 1.27 | 1.42 | 1.07 | 0.99 | 0.87 | 0.54 | 1.15 | 1.18 |
| SparseCraft | 1.04 | 1.17 | 1.74 | 1.80 | **0.70** | 1.19 | 1.53 | 0.83 | **1.05** | 1.42 | 0.78 | **0.80** | 0.56 | **0.44** | **0.77** | **0.84** |
| PixelNeRF | 6.18 | 5.13 | 8.07 | 5.85 | 4.40 | 7.11 | 4.64 | 5.68 | 6.76 | 9.05 | 6.11 | 3.95 | 5.92 | 6.26 | 6.89 | 6.93 |
| IBRNet | 2.32 | 2.29 | 3.70 | 2.66 | 1.83 | 3.02 | 2.83 | 1.77 | 2.28 | 2.73 | 1.96 | 1.87 | 2.13 | 1.58 | 2.05 | 2.09 |
| MVSNeRF | 2.09 | 1.96 | 3.27 | 2.54 | 1.93 | 2.57 | 2.71 | 1.82 | 1.72 | 2.29 | 1.75 | 1.72 | 1.47 | 1.29 | 2.09 | 2.26 |
| SparseNeuS | 1.96 | 2.17 | 3.29 | 2.74 | 1.67 | 2.69 | 2.42 | 1.58 | 1.86 | 1.94 | 1.35 | 1.50 | 1.45 | 0.98 | 1.86 | 1.87 |
| VolRecon | 1.38 | 1.20 | 2.59 | 1.56 | 1.08 | 1.43 | 1.92 | 1.11 | 1.48 | 1.42 | 1.05 | 1.19 | 1.38 | 0.74 | 1.23 | 1.27 |
| ReTR | 1.17 | 1.05 | 2.31 | 1.50 | 0.96 | 1.20 | 1.54 | 0.89 | 1.34 | 1.30 | 0.87 | 1.06 | 0.77 | 0.59 | 1.06 | 1.11 |
| C2F2NeuS | 1.11 | 1.12 | 2.42 | 1.40 | 0.75 | 1.41 | 1.77 | 0.85 | 1.16 | 1.26 | 0.76 | 0.91 | 0.60 | 0.46 | 0.88 | 0.92 |
| UFORecon | **1.00** | 0.79 | 2.03 | **1.33** | 0.87 | **1.11** | **1.19** | 0.74 | 1.22 | 1.14 | **0.71** | 0.89 | 0.59 | 0.56 | 0.90 | 1.02 |
| *SVRecon* | **1.00** | **0.72** | 1.98 | 1.44 | 0.83 | 1.12 | 1.40 | 0.72 | 1.27 | **1.06** | 0.76 | 0.94 | 0.56 | **0.44** | 0.87 | 0.93 |

Table 2: **Quantitative evaluation results of sparse-view reconstructions on 15 testing scenes of DTU dataset using *Chamfer Distances*.** We test and report results using released codes for VolRecon, ReTR, and UFORecon, other results are sourced from these papers. From top to bottom, the baseline methods are from different categories: 1) Multi-view Stereo (MVS) methods; 2) neural implicit reconstruction methods (requiring per-scene optimization); 3) generalizable neural rendering methods; 4) generalizable surface reconstruction methods. We use **bold** to indicate best performance, and underline to indicate the second-best. ReTR is the closest baseline and also the one we built our approach upon.

| Scan | AUC@15°↑ | 24 | 37 | 40 | 55 | 63 | 65 | 69 | 83 | 97 | 105 | 106 | 110 | 114 | 118 | 122 |
|---|---|---|---|---|---|---|---|---|---|---|---|---|---|---|---|---|
| VolRecon | 8.4 | 8.3 | 7.0 | 14.2 | 12.1 | 5.7 | 8.2 | 7.8 | 5.0 | 5.8 | 6.6 | 10.8 | 3.8 | 11.9 | 8.5 | 10.5 |
| ReTR | 16.3 | 20.0 | 11.5 | 26.3 | 20.5 | 14.7 | 14.8 | 15.7 | 9.4 | 12.7 | 12.1 | 19.9 | 14.4 | 21.8 | 15.2 | 15.4 |
| UFORecon | 11.8 | 14.6 | 8.3 | 14.4 | 15.8 | 8.1 | 11.3 | 11.8 | 6.5 | 7.5 | 9.6 | 16.4 | 10.4 | 15.6 | 13.5 | 13.9 |
| *SVRecon* | **21.0** | **26.3** | **14.6** | **28.8** | **26.4** | **18.5** | **18.1** | **20.3** | **11.4** | **15.3** | **14.9** | **26.3** | **19.6** | **29.3** | **22.2** | **22.9** |

Table 3: **Quantitative evaluation results of sparse-view reconstructions on 15 testing scenes of DTU dataset using *Normal Consistency*.** We test and report results using released codes from the compared methods. We use **bold** to indicate best performance.

## 5 EXPERIMENTAL RESULTS

In this section, we begin by outlining our experimental setup, including datasets and implementation details. Next, we assess our approach both qualitatively and quantitatively on generalizable surface reconstruction against state-of-the-art competitors. We also provide evaluation results to demonstrate the effectiveness of our occupancy prediction network. We finally present results to evaluate the generalization ability of our method without retraining and conduct analyses for different elements in our method.

**Datasets.** As in previous works (Liang et al., 2024; Ren et al., 2023; Xu et al., 2023; Na et al., 2024), we use the DTU (Aanæs et al., 2016) dataset for training and main evaluation. It consists of high-resolution images of 124 different scenes under 7 lighting conditions captured under controlled laboratory conditions, each accompanied by accurate camera matrix and laser-scanned ground truth depth map. We use the same evaluation protocol as in earlier work and 3 views as input for each one of the 15 test scenes. In addition to DTU, we also use BlendedMVS (Yao et al., 2020) and Tanks and Temples (Knapitsch et al., 2017) dataset to evaluate the generality of our approach.

**Implementation Details.** We use $M = 4$ input views with resolution $640 \times 512$ in training, and $M = 3$ input views with resolution $800 \times 600$ in testing, consistent with previous works. In volumetric feature construction Sec. 4.1, we use $C = 32$ feature channels. Our model is trained for 16 epochs using Adam optimizer (Kingma & Ba, 2014) the learning rate is set to $10^{-4}$. Please refer to Supp. A.2 for more details.

### 5.1 COMPARATIVE RESULTS ON DTU

We first present our evaluation metrics and the baselines we compare against. We then report quantitative results in Tab. 2 and Tab. 3, along with qualitative ones in Fig. 3.

| Scan | Mean | 24 | 37 | 40 | 55 | 63 | 65 | 69 | 83 | 97 | 105 | 106 | 110 | 114 | 118 | 122 |
|---|---|---|---|---|---|---|---|---|---|---|---|---|---|---|---|---|
| Precision | 23.0 | 28.0 | 26.2 | 26.1 | 23.9 | 26.7 | 26.6 | 23.6 | 24.7 | 29.0 | 26.0 | 22.5 | 13.2 | 18.9 | 16.4 | 13.3 |
| Recall | 96.8 | 98.5 | 91.2 | 90.4 | 98.4 | 92.4 | 94.7 | 99.1 | 97.0 | 98.1 | 97.0 | 97.7 | 98.3 | 99.8 | 99.3 | 99.9 |
| Space Occupation (Ours) | 1.89 | 2.00 | 2.06 | 1.80 | 1.63 | 2.13 | 1.57 | 1.57 | 2.35 | 2.02 | 2.24 | 1.58 | 1.96 | 1.55 | 1.91 | 2.01 |
| Space Occupation (GT) | 0.45 | 0.57 | 0.59 | 0.52 | 0.39 | 0.62 | 0.44 | 0.37 | 0.60 | 0.60 | 0.60 | 0.36 | 0.26 | 0.29 | 0.32 | 0.27 |

Table 4: **Evaluation results of occupancy predictions on 15 testing scenes of `DTU` dataset.** We use precision and recall to quantify the performance of occupancy prediction. We also provide space efficiency statistics defined as the ratio between number of occupied voxels and number of all voxels. All reported results are in percentage.

**Evaluation Metrics.**    *Chamfer Distances* between predicted surfaces and ground truth point clouds have been extensively used and is a standard metric to evaluate surface reconstruction quality. Unfortunately, it lacks awareness of local geometry and density and as a result, does not accurately quantify proper recovery of fine-scale details and reconstructed surface smoothness. Thus, we also report a *Normal Consistency* metric that accounts for surface quality. To evaluate it, we first extract 3D meshes from predicted depth maps and ground truth depth maps using TSDF fusion (Curless & Levoy, 1996) and Marching Cube (Lorensen & Cline, 1987). We then compute the angular differences between normals at closest vertices in the two meshes, and use Area Under the Curve (AUC) up to $15°$ in percentage to measure the overall normal consistency robustly. In Fig. 5 in Supp., we show the angular differences on a specific example.

**Results.**    As reported in Tabs. 2 and 3, in terms of Chamfer distance, our method is on par with UFORecon and they both outperform all other methods. However, as can be clearly seen in Fig. 3, the surfaces produced by UFORecon are much rougher than ours. We introduce the normal consistency metric to measure this qualitative but important difference, since Chamfer distance alone does not capture that property. In terms of that metric, we do much better than UFORecon because, by using a high-resolution volumetric representation, we get much more regular and smoother surfaces. ReTR is the closest baseline to us in methodology and we clearly do better on all metrics and for all scenes. In Fig. 3, this manifests itself by the fact that our reconstructed surfaces are much smoother than UFORecon and devoid of small artifacts that ReTR creates.

## 5.2    Occupancy Prediction

Occupancy prediction is the first stage in our method and the second stage depends on it being accurate. We therefore evaluate it by itself.

**Evaluation Metrics.**    In essence, it is a classification problem for which *Precision* and *Recall* can be used as evaluation metrics. To quantify the savings of our sparse feature volumes scheme in storage, we also compute a *Space Occupation* metric, taken to be the ratio between number of occupied voxels and number of all voxels.

There is in general a trade-off between precision and recall, however in our surface reconstruction problem, we prioritize recall and compromise precision such that surface geometry is preserved as much as possible. In practice, 1) we use a conservative threshold of $0.1$ to determine the occupied voxels from the occupancy prediction from the network $\mathbf{O}$; 2) we then further dilate the occupied voxels using a cubic $3 \times 3 \times 3$ kernel to obtain the final occupancy prediction results.

**Results.**    We report our quantitative in Tab. 4, and provided qualitative ones in Fig. 3. We achieve a $96.8\%$ average *Recall*, ensuring the preservation of surface geometry. The false negatives mostly come from the textureless table in the scenes, which is very hard to reconstruct and does not take part in the standard evaluation. The *Precision* is on average at $23.0\%$, resulting in *Space Occupation* at $1.89\%$, $4.2$ times of the optimal *Space Occupation* at $0.45\%$. This demonstrates the strong performance of our occupancy prediction method, which recalls most of the surface by keeping only $1.89\%$ of the voxels on average.

## 5.3    Further Analyses

**Memory Consumption.**    We consider only the second stage here, as the first stage of occupancy prediction can be run seperately beforehand and does not consume much memory. Practically, our method consumes around 30 GB memory to train with batch size as 1, 1024 sampled rays and 32 feature channels in the second stage. This is a moderate requirement for modern GPUs, but also

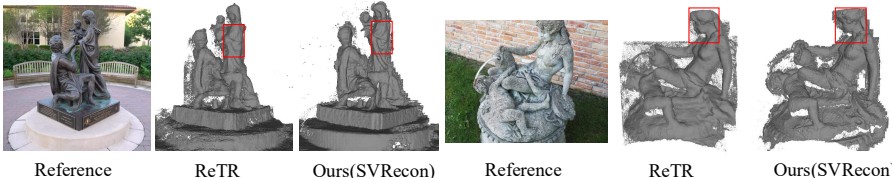

Reference  ReTR  Ours(SVRecon)  Reference  ReTR  Ours(SVRecon)

Figure 4: We apply our method, pretrained only on the `DTU` dataset, to scenes from `Tanks and Temples` (top) and `BlendedMVS` (bottom) datasets. The high-quality reconstructed surfaces highlight the strong generalization ability of our method. The red rectangles highlight the detail-preserving ability of our method.

prevents us from lifting the resolution even more. In testing, the memory requirment is eased even more to around 12 GB with the same setting.

**Impact of Different Resolutions.** The resolution is of critical importance in the task of surface reconstruction. To verify this point, we use the same occupancy prediction results as in our main experiment and vary the times of supersampling in the second stage of our method. We test our method at 1, 2 times of supersampling, leading to resolutions at $128^3$ and $256^3$. Compared to the adopted resolution in our method at $512^3$, the tested resolutions are only lower, because we cannot lift the resolution even more due to memory consumption constraints. The results are presented in Tab. 5. As we can see, there is an abrupt improvement in performance from $128^3$ to $256^3$, and a mild improvement from $256^3$ to $512^3$. Overall, this validates the intuitition that the performance will increase as the resolution increases.

**Number of Views.** While our method is trained on $M = 4$ input views, it is not restricted to this setting due to the mean and variance feature construction operation as in section 4.1. In addition to the $M = 3$ setting in our main experiment, We also test the performance of our method with $M = 4$ and $M = 5$ input views, and the results are given in table 5. It can be observed that the surface reconstruction quality increases with more input views, as more scene information becomes available.

| Settings | Mean (CD) ↓ |
|---|---|
| Resolution @ $128^3$ | 1.27 |
| Resolution @ $256^3$ | 1.04 |
| Number of Views @ 5 | 0.96 |
| Number of Views @ 4 | 0.99 |
| Base Feature Channels @ 16 | 1.15 |
| Ours ($SVRecon$) | 1.00 |

Table 5: A study on the impact of different settings on the final performance of our method, including **resolution**, **number of input views** and **number of base feature channels**.

**Number of Base Feature Channels.** The number of feature channels is crucial in determining the effectiveness of scene feature representations. However, more feature channels will also incur more memory burden. In our main experiment, we use $C_f = 32$ base feature channels in volumetric feature construction. We also test the performance of our method with $C_f = 16$ base feature channels, and the results are provided in table 5. It shows that reducing the number of base channels will degrade the performance.

**Generalization Ability.** To validate the generalization ability of our method in surface reconstruction, we apply our method, pretrained only on `DTU` dataset, to scenes from `BlendedMVS` (Yao et al., 2020) and `Tanks and Temples` (Knapitsch et al., 2017). In this experiment, we use $M = 5$ input views, the visualization is given in Fig. 4. It can be seen that our method generates surfaces with finer details than ReTR, demonstrating the strong generalization ability.

## 6 CONCLUSION AND FUTURE WORK

In this paper, we propose a two-stage neural surface reconstruction method based on the efficient scene representation of sparse feature volumes. In the first stage, our method is capable of performing accurate occupancy prediction, retaining only around 1.9% of all voxels as occupied voxels and greatly reducing the memory burden. In the second stage, our method can reconstruct high-quality surfaces by conducting feature-based volume rendering on the constructed sparse feature volumes at a high resolution $512^3$. Extensive experiments have demonstrated the superiority of our method compated to a variety of existing methods in terms of surface reconstruction quality. In the future, we will explore more efficient schemes that generalizes to realistic unbounded scenes with arbitrary number of views.

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

# A APPENDIX

## A.1 QUERYING SPARSE VOLUMES.

Hereafter, we will use *coarse-voxel* to indicate a voxel at occupancy prediction resolution $K^3$, and *fine-voxel* to indicate a mini-voxel in each mini-volume effectively at resolution $(sK)^3$. To explain the algorithm, we first define a global grid frame, an occupancy grid frame and a local grid frame. The global grid frame is a hypothetical one at the resolution $(sK)^3$, corresponding to the densified sparse feature volume and each vertice representing a *fine-voxel* center point. The occupancy grid frame is at the resolution of occupancy prediction $K^3$, each vertice representing a *coarse-voxel* center point. Each occupied voxel spans a mini-volume with a local grid frame at resolution $s^3$, where each vertice represents a *fine-voxel* center point as well. For an arbitrary query location $\mathbf{p}$, we need to get the associated features of its eight adjacent vertices from S to perform trilinear interpolation to obtain $\mathbf{p}$'s feature representation. We detail on this functionality in the following.

Knowing the size of a *coarse-voxel*, it is straightforward to find $\mathbf{p}$'s nearest vertices in global grid frame and compute their global coordinates $\{v_g \mid v_g \in \{0, 1, ..., sK - 1\}^3\}$. Knowing the supersampling rate $s$, we can easily convert the global coordinates into occupancy coordinates $\{v_o \mid v_o \in \{0, 1, ..., K-1\}^3\}$ and local grid coordinates $\{v_l \mid v_l \in \{0, 1, ..., S-1\}^3\}$. Now we consider the data structure of sparse feature volumes $\mathsf{S} \in \mathbb{R}^{N \times C \times s^3}$. To query the features associated with vertices, we can use directly local grid coordinates for indexing in the last three dimensions. The problem now reduces to finding the right mini-volme index in the first dimension from occupancy coordinates, for which we propose to use a mapping function $\mathbb{H} : \{0, 1, ..., K - 1\}^3 \rightarrow \mathbb{Z}^+$ to map $v_o$ to the sought index $n$. For that purpose, we define a regular tensor as the dense lookup table that encodes mapped values in its entries. The tensor $\mathbb{H} \in \mathbb{R}^{K^3}$ is at a low resolution, initialized with values of -1 that points to a dummy feature. Then we simply apply boolean indexing in `PyTorch` to encode mini-volume indexes, i.e. $\mathbf{H}[\mathbf{O}] = \{0, 1, ..., N - 1\}$. The boolean indexing is consistent with S in its creation, such that $n = \mathbf{H}[v_o]$ can be used to index S in the first dimension. Having the associated features of the eight adjacent vertices, trilinear interpolation can be performed for the query result, which completes the query process. Given a 3D query point $\mathbf{p}$ with coordinates $[x_p, y_p, z_p]$ in a sparse scene containing N voxels $\{v_1, ..., v_N\}$, we need to determine the index $i \in \{1, ..., N\}$ of the sparse voxel $v_i$ containing $\mathbf{p}$. To this end, we construct a 3D hash table $\mathbf{H}$ such that $\mathbf{H}(x_p, y_p, z_p) = i$, mapping spatial coordinates to their corresponding voxel indices. It can be done by performing boolean indexing on an 3D array with "False" values everywhere except for each position of the sparse voxels.

## A.2 MORE IMPLEMENTATION DETAILS.

**Method Configurations.** For occupancy prediction, the voxel grid resolution is set to $128^3$. After binarizing the initial predictions using threshold $\tau = 0.1$, a morphological dilation process is further applied to the prediction results to maximize the recall of geometry. In the second stage, we supersample each occupied voxel by $s = 4$ times in our experiments, leading to resolutions at $512^3$.

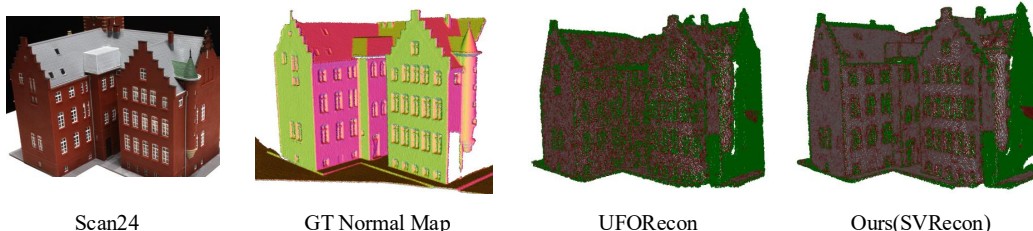

|  |  |  |  |
|---|---|---|---|
| Scan24 | GT Normal Map | UFORecon | Ours(SVRecon) |

Figure 5: *Normal Consistency* **evaluation on scan24 from the `DTU` dataset.** The normal differences are visualized using colors. Errors in the range $[0°, 15°]$ are color coded linearly from white to red. Error larger than $15°$ are shown in green. Our $SVRecon$ method significantly outperforms UFORecon in *Normal Consistency*.

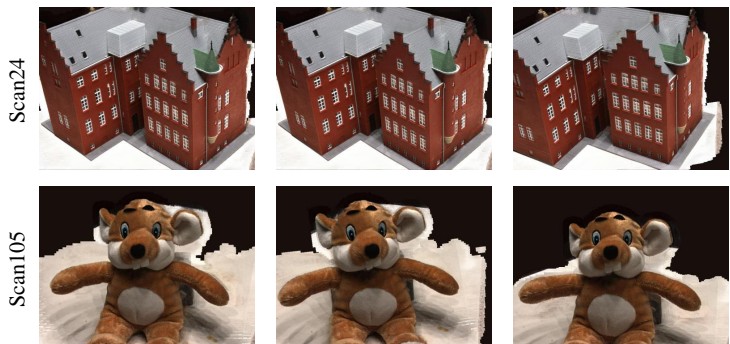

Figure 6: Novel view synthesis examples of our $SVRecon$ method on scan24 and scan105 from the `DTU` dataset. The details are preserved well for the foreground object.

In ray sampling, we sample 64 points per-ray both in training and testing. Our model is trained on 4 A100 GPUs for 16 epochs. To reconstruct the surface, we follow the existing works (Liang et al., 2024; Ren et al., 2023; Xu et al., 2023; Na et al., 2024) to define a virtual rendering viewpoint corresponding to each view by shifting the original camera coordinate frame by $25mm$ along its $x$-axis, and then use TSDF fusion (Curless & Levoy, 1996) to merge the rendered depth maps in a volume and extract the mesh from it using Marching Cube (Lorensen & Cline, 1987).

**Network Architectures.** We use Feature Pyramid Network (FPN) for image feature extraction. To obtain projection features from 3D points, we interpolate on the $1/2$ resolution feature map from FPN. For 3D sparse feature volumes we use a sparse UNet architecture, with 4 encoder layers at dimension $[32, 64, 128, 256]$, bottleneck layer at dimension $256$, and decoder layers at dimension $[256, 128, 64, 32]$.

**Dilation after Network Occupancy Prediction.** To maximize recall, we first use a thresold $\tau = 0.1$ to binarize sthe occupancy predictions from the network. Then we apply a dilation process on the binary occupancy fields. In the dilation process, we apply convolution with a $3 \times 3 \times 3$ kernel on the occupancy fields to compute a score for each voxel, and then set a threshold $3^3 * 0.1$ to binarize the occupancy fields again as the final occupancy prediction results. By doing this, more voxels near the originally predicted voxels are included to improve recall. The process is written as $\widetilde{\mathbf{O}} = \text{dilate}(\mathbb{I}(\mathbf{O} \geq \tau))$.

### A.3 More Visulizations.

**Normal Consistency Visualization.** We visualize an example of normal consistency evaluation in Fig. 5.

**Novel View Synthesis.** Our method can also perform novel view syntheiss using the trained volume rendering network, some visual examples are presented in Fig. 6. Due to our sparse representation, the background is not modeled in the results.

