# OpenReview forum: "High-Fidelity and Generalizable Neural Surface Reconstruction with Sparse Scene Representations"
_ICLR.cc/2026/Conference — ICLR 2026 Conference Withdrawn Submission_

### Official Review · Reviewer_tXND · 2025-10-26

**Soundness:** 3
**Presentation:** 3
**Contribution:** 3
**Rating:** 4
**Confidence:** 4

**Summary:**

## Summary
This paper introduces SVRecon, a novel method for high-fidelity generalizable neural surface reconstruction from sparse input views with camera poses. Unlike previous generalizable neural surface reconstruction (GNSR) methods that rely on dense 3D feature volumes—which are memory-intensive and limit resolution—SVRecon adopts a sparse scene representation to enable efficient and high-resolution reconstruction.

**Strengths:**

## Strength

SVRecon achieves high-fidelity surface reconstruction from only a few input views without per-scene optimization, and outperforms existing generalizable methods on the DTU dataset in both resolution and accuracy.
The design of the sparse volume representation is well-motivated and effectively addresses memory constraints.
The paper is clearly written and easy to follow.

**Weaknesses:**

## Weakness
1. Novelty of Sparse Representation: Sparse representations have been explored in other 3D reconstruction works such as Trellis and Sparc3D. The authors should more clearly differentiate their specific design choices and contributions relative to these existing methods.

2. Limited Training Data: The model is trained only on the DTU dataset, raising concerns about its generalization capability. Training on a larger and more diverse dataset (e.g., Objaverse) and reporting results on additional benchmarks would strengthen the validity of the claims.

**Questions:**

Since the first stage predicts occupancy, it would be valuable to include qualitative results on full object reconstruction to better assess the generalization ability of the method beyond the DTU test scenes.

---

### Official Review · Reviewer_vQFB · 2025-10-31

**Soundness:** 3
**Presentation:** 2
**Contribution:** 2
**Rating:** 4
**Confidence:** 4

**Summary:**

**Overview**

This paper proposes a method for improving generalizable neural surface reconstruction using a sparse voxel representation. The work addresses an important research direction in 3D reconstruction by combining neural fields with hybrid implicit/voxel representations to achieve higher resolution reconstructions. While neural surface reconstruction has been extensively studied, the push toward generalizable approaches with sparse scene representations represents a relevant contribution to the field.

**Recommendation**

This paper is **marginally below the acceptance threshold**. While it tackles an important problem and demonstrates promising results, several critical issues prevent acceptance:

- Presentation quality: Mathematical notation inconsistencies (inconsistent indexing, symbol reuse, ambiguous conventions) severely hinder readability and reproducibility.

- Incomplete evaluation: Quantitative results are only on in-domain DTU. The absence of quantitative results on BlendedMVS and Tanks & Temples, plus missing comparisons to recent non-neural surface-based baselines (e.g. Dust3R [9], MASt3R [10], AnySplat [11], VGGT [12]), makes it difficult to assess the true generalization capability.

- Incomplete related work: The historical account starts with NeRF while overlooking foundational works (Occupancy Networks [1], DeepSDF [2], Scene Representation Networks [3]).

- Missing analysis: Lacks ablation studies and memory analysis for understanding trade-offs.

However, the problem is relevant, the motivation is well-articulated, and the DTU results are promising. With thorough revisions, this paper could make a solid contribution.

**Strengths:**

The paper has several notable qualities:

1. Clear motivation: The paper is well-structured with a clear motivation for the proposed approach. The rationale for using sparse voxel representations in generalizable neural surface reconstruction is well-articulated.

2. Simple approach: The two-stage design with a low-resolution occupancy predictor followed by a sparse UNet is simple and makes sense. The description of ray sampling and querying for sparse volumes is a meaningful contribution.

3. Promising results: The experimental results demonstrate improvements over baseline methods, showing the potential of the proposed approach.

4. Important research direction: Generalizable neural scene prediction is a highly relevant research topic with broad applications in computer vision and graphics.

**Weaknesses:**

**Presentation Issues**

The paper suffers from significant inconsistencies in mathematical notation that hinder readability:

- Image notation: Images are denoted inconsistently as *I_i* in some places and *C* in others (l. 151, Eqs. 2-4, l. 175)
- Bold notation: Bold symbols are used inconsistently (e.g. image *I* is sometimes bold, sometimes not). In Appendix A.1, **H** and **H** (with different font styles) appear to be used in the same way. Generally, there seems to not be a consistent convention when to use what font style.
- Ground truth notation: No consistent convention to distinguish predictions from ground truth (*Ĉ* vs *C_g* in Eqs. 3-4 vs Eq. 10)
- Inconsistent indexing: In Section 1, the index *i* is used ambiguously for both query points and images (l. 151 and Eq. 1). In Appendix A.1, *i* and *n* appear to be used interchangeably.
- Symbol reuse:
  - *M* is used both as an index (l. 151) and for depth maps (l. 170). The latter is inconsistent to the usage of D for depth maps (e.g. Eq. 4).
  - *N_s* and *N* appear in Eq. 9, but Eq. 10 introduces *N_s* and *N_d* for sampled pixels. Additionally, *N* also denotes the number of images (l. 151) and the number of occupied voxels (l. 250 and Appendix A.1)
  - *D* represents both depth maps and dimensions (e.g., Eq. 9)
  - *O* and *Õ* are used inconsistently for both "occupancy probability" (e.g., l. 696) and "binary occupancy" (e.g., l. 634)

There are likely more issues and I highly recommend that the authors proofread the paper to ensure that the list of symbols is consistent.

**Experimental Evaluation**

The experimental validation, while showing promise, has several gaps that limit the assessment of the method's generalization capabilities:

1. Limited quantitative evaluation: Quantitative results are only provided on DTU (which is in-domain), and even on DTU, the results are not consistently better than UFORecon in terms of Chamfer Distance. The paper lacks quantitative results on BlendedMVS and Tanks & Temples, which would better demonstrate generalization capabilities.

2. Metric definitions: The normal consistency metric appears to be defined arbitrarily. The authors should use definitions and hyperparameters from established prior work (with proper citations) to ensure fair comparisons.

3. Insufficient qualitative comparisons: The paper provides very few qualitative comparisons. Detailed qualitative results on DTU, BlendedMVS, and Tanks & Temples should be added to the supplementary material.

4. Missing baselines: The paper does not compare against recent non-neural surface-based methods such as Dust3R [9], MASt3R [10], AnySplat [11], and VGGT [12].

**Missing Ablations and Analysis**

The following analyses would strengthen the paper:

1. Memory breakdown: Show how the 12 GB inference memory is allocated across different network components
2. Scene-dependent memory: Analyze how memory consumption varies between different scenes
3. Threshold ablation: Provide an ablation on the threshold τ for voxel pruning, including a memory/accuracy trade-off curve

**Minor Points**

- Figure 4 caption refers to "top and bottom" but only shows left/right images
- Many works use SSIM loss in addition to L1 for photometric supervision (Eq. 3). Consider discussing this design choice.
- Typos: "accuraciy" (l. 16), "requirment" (l. 444), and "syntheiss" (l. 702). Please run a spell-check.
- The motivation for using voxels over alternative representations could be more explicitly stated early in the paper.

**Related Work**

The related work section presents an incomplete account of the historical development of the field. The progression should be presented as follows: Occupancy Networks [1] and DeepSDF [2] were the pioneering works in implicit neural representations. Scene Representation Networks [3] built upon these foundations and, together with DVR [4] and its successor IDR [5], inspired NeRF [6]. Subsequently, methods like NeuS [7] and Convolutional Occupancy Networks [8] built upon these foundations. The narrative should not start with NeRF, as implicit surface representations preceded it.

**References**

[1] Mescheder et al.: Occupancy Networks: Learning 3D Reconstruction in Function Space (CVPR 2019)

[2] Park et al.: DeepSDF: Learning Continuous Signed Distance Functions for Shape Representation (CVPR 2019)

[3] Sitzmann et al.: Scene Representation Networks: Continuous 3D-Structure-Aware Neural Scene Representations (NeurIPS 2019)

[4] Niemeyer et al.: Differentiable Volumetric Rendering: Learning Implicit 3D Representations without 3D Supervision (CVPR 2020)

[5] Yariv et al.: Multiview Neural Surface Reconstruction by Disentangling Geometry and Appearance (NeurIPS 2020)

[6] Mildenhall et al.: NeRF: Representing Scenes as Neural Radiance Fields for View Synthesis (ECCV 2020)

[7] Wang et al.: NeuS: Learning Neural Implicit Surfaces by Volume Rendering for Multi-View Reconstruction (NeurIPS 2021)

[8] Peng et al.: Convolutional Occupancy Networks (ECCV 2020)

[9] Wang et al.: DUSt3R: Geometric 3D Vision Made Easy (CVPR 2024)

[10] Leroy et al.: Grounding Image Matching in 3D with MASt3R (arXiv 2024)

[11] Jiang et al.: AnySplat: Feed-forward 3D Gaussian Splatting from Unconstrained Views (SIGGRAPH Asia 2025, ACM TOG)

[12] Wang et al.: VGGT: Visual Geometry Grounded Transformer (CVPR 2025)

**Questions:**

The following questions would help clarify the method and its evaluation:

1. How was the train/test split on DTU determined?
2. L. 308: How is feature aggregation performed specifically?
3. How is the scene bounding box determined (l. 216)?
4. Are the low-resolution occupancy predictor and high-resolution sparse UNet trained end-to-end or in two stages? How do the authors ensure that the features from the low-resolution model are useful to the sparse high-resolution model?
5. Can the authors analyze error accumulation in the two-step reconstruction process? How critical is the first step to overall performance?

---

### Official Review · Reviewer_p3Ft · 2025-10-31

**Soundness:** 1
**Presentation:** 2
**Contribution:** 1
**Rating:** 2
**Confidence:** 5

**Summary:**

This paper proposed a generalizable method to reconstruct the neural surface. They desinged the model based on the feature volume and proposed a two-stage solution to mitigate the huge memory overhead of the classical dense volume.

**Strengths:**

1. The overall presentation is clear.
2. the motivation of sparsifing the volume to save memory overhead is reasonable.

**Weaknesses:**

1. The main ideal of this paper to sparsify the volume to save memory overhead is very common and has been adopted by many existing methods like SparseNeus and SuRF [1] (its citation is also missing).
2. Existing methods (SparseNeus and SuRF) can already sparsify the volume unsupervised, but this paper further needs the ground-true depth, which harms the simplicity and convenience. SuRF can even achieve end-to-end training of multiple stages, but this paper still needs two-stage seperately training and will inevitably introduces accumulated error.
3. The interpolation on the sparse volume based on the query table is also not the new algorithm which has been proposed by SuRF [1] and HIVE [2].
4. There is only one scene on BlendedMVS and Tanks&Temples comparison experiment, and needs more comparison on complex scenes to prove the generalizability.


[1] Surface-Centric Modeling for High-Fidelity Generalizable Neural Surface Reconstruction, ECCV2024.
[2] HIVE: HIerarchical Volume Encoding for Neural Implicit Surface Reconstruction.

**Questions:**

This paper has very limited novelty. The adopted sparsification strategy is a very common strategy in generalizable neural surface reconstruction field, and the designed supervised two-stages pipeline is even the outdated strategies.

---

### Note · Authors · 2025-11-13

I have read and agree with the venue's withdrawal policy on behalf of myself and my co-authors.